# A Novel Single-Stranded RNA-Based Adjuvant Improves the Immunogenicity of the SARS-CoV-2 Recombinant Protein Vaccine

**DOI:** 10.3390/v14091854

**Published:** 2022-08-24

**Authors:** Dong Liu, Chaoqiang An, Yu Bai, Kelei Li, Jianyang Liu, Qian Wang, Qian He, Ziyang Song, Jialu Zhang, Lifang Song, Bopei Cui, Qunying Mao, Wei Jiang, Zhenglun Liang

**Affiliations:** 1Division of Hepatitis and Enterovirus Vaccines, NHC Key Laboratory of Research on Quality and Standardization of Biotech Products, NMPA Key Laboratory for Quality Research and Evaluation of Biological Products, Institute of Biological Products, National Institutes for Food and Drug Control, Beijing 102600, China; 2Changchun Institute of Biological Products Co., Ltd., Changchun 130062, China; 3Beijing Minhai Biotechnology Co., Ltd., Beijing 102629, China

**Keywords:** single-stranded RNA-based adjuvant, immune response, SARS-CoV-2 recombinant protein vaccine

## Abstract

The research and development (R&D) of novel adjuvants is an effective measure for improving the immunogenicity of the severe acute respiratory syndrome coronavirus 2 (SARS-CoV-2) recombinant protein vaccine. Toward this end, we designed a novel single-stranded RNA-based adjuvant, L2, from the SARS-CoV-2 prototype genome. L2 could initiate retinoic acid-inducible gene-I signaling pathways to effectively activate the innate immunity. ZF2001, an aluminum hydroxide (Al) adjuvanted SARS-CoV-2 recombinant receptor binding domain (RBD) subunit vaccine with emergency use authorization in China, was used for comparison. L2, with adjuvant compatibility with RBD, elevated the antibody response to a level more than that achieved with Al, CpG 7909, or poly(I:C) as adjuvants in mice. L2 plus Al with composite adjuvant compatibility with RBD markedly improved the immunogenicity of ZF2001; in particular, neutralizing antibody titers increased by about 44-fold for Omicron, and the combination also induced higher levels of antibodies than CpG 7909/poly(I:C) plus Al in mice. Moreover, L2 and L2 plus Al effectively improved the Th1 immune response, rather than the Th2 immune response. Taken together, L2, used as an adjuvant, enhanced the immune response of the SARS-CoV-2 recombinant RBD protein vaccine in mice. These findings should provide a basis for the R&D of novel RNA-based adjuvants.

## 1. Introduction

The coronavirus disease 2019 (COVID-19), caused by severe acute respiratory syndrome coronavirus 2 (SARS-CoV-2), has led to a global pandemic with more than 500 million confirmed cases and more than 6 million deaths as of June 2022, as reported by the World Health Organization (WHO) [1]. To combat the COVID-19 pandemic, vaccines were researched and developed successfully employing different technical approaches. As of date, more than 11 billion doses of vaccines have been used for immunizing human beings worldwide [1]. Of these, billions of doses of inactivated virus vaccines and recombinant protein vaccines have been used for immunization in developing countries, such as China, which has played a critical role in preventing severe illness and reducing hospital admissions [2]. Frustratingly, the emergence of SARS-CoV-2 variants, especially variants of concern (VOCs) defined by the WHO, has lowered the efficacy of most vaccines to varying degrees [3]. Thus, one of the priority measures against variants, such as the Omicron variant, with higher transmission ability, is to improve the vaccine immunogenicity. It is believed that optimizing the antigen design is a feasible approach to improving the immunogenicity. For example, the receptor binding domain (RBD)-dimer, as a tandem-repeat single-chain, was subtly designed in ZF2001, a vaccine with emergency use authorization in China, to improve the immunogenicity compared with that of the monomer RBD [4,5]. Subsequently, the Delta-Omicron RBD-dimer was designed as an antigen to prepare the second-generation recombinant protein vaccine to combat the epidemics of the Delta and Omicron variants [6]. In addition, due to the relatively lower immunogenicity of inactivated virus vaccines and recombinant protein vaccines, it is imperative that the vaccines have compatibility with adjuvants that can effectively improve their immunogenicity. Despite intensive efforts for almost 100 years, only a few adjuvants have been approved. At present, most of the inactivated virus vaccines and recombinant protein vaccines use aluminum hydroxide (Al) as an adjuvant [7]. Because of the poor persistence of SARS-CoV-2 vaccines and the immune evasion of VOCs, there is an urgent need to undertake research and develop novel vaccine adjuvants with stronger effects to meet the demand. Some novel adjuvants, such as MF59, Matrix-M^TM^, and AS01, have successively moved to clinical trials worldwide [8,9].

Activation of the innate immunity is one of the mechanisms underlying the function of vaccine adjuvants [10,11]. The initiation of pattern recognition receptor (PRR) signaling pathways, such as retinoic acid-inducible gene-I (RIG-I), nucleotide-binding oligomerization domain 2 (NOD2), and Toll-like receptor (TLR) signaling pathways, is a key step in the activation of innate immunity [12,13]. Therefore, ligands of PRRs have potential ability as vaccine adjuvants. Recently, CpG 7909, the ligand for TLR-9, was used as an adjuvant in a hepatitis B vaccine in clinical trials [14]. The compatibility of an inactivated subunit influenza vaccine with a novel single-stranded RNA (ssRNA) originating from influenza HA2 genes through the activation of PRR signaling pathways was shown, and could induce higher titers of neutralizing antibodies as well as a balanced Th1/ Th2 response in mice [15,16]. It could also provide effective protection in aged mice [17].

Numerous studies have indicated that the long ssRNA sequence containing a hairpin structure in the SARS-CoV-2 genome could be recognized by RIG-I in individuals post infection [18,19,20]. Based on this knowledge, we designed an ssRNA sequence, L2, originating from the SARS-CoV-2 prototype 5′-untranslated region (UTR), 3′-UTR, and RBD genome, as an adjuvant, compatible with the RBD peptide to prepare a SARS-CoV-2 recombinant protein vaccine and used it to immunize mice. We evaluated whether L2 was more effective in improving the immunogenicity of the vaccine compared with Al and other nucleic acid adjuvants targeted for PPRs, such as CpG 7909 and double-stranded polyriboinosinic:polyribocytidylic acid (poly(I:C)), in mice.

## 2. Materials and Methods

### 2.1. Biosafety and Ethics Statement

All experiments involving live SARS-CoV-2 were performed in a biosafety Level 3 laboratory (ASBL-3) of Sinovac Life Sciences Co., Ltd., Beijing, China. All animal studies were conducted in accordance with relevant ethical regulations and were approved by the institutional animal care and use committee of National Institutes for Food and Drug Control (Approval number: R-NIFDC-PD-301-05).

### 2.2. Preparation of RNA L1 (L1) and RNA L2 (L2)

L2 was designed based on the 5′-UTR, 3′-UTR, and RBD of the SARS-CoV-2 prototype genome with bGH poly A or without bGH polyA (L2-no bGH poly A). bGH poly A signal is often used for plasmid vectors. L1 was designed based on the 5′- and 3′-UTR of the prototype genome with bGH poly A or without bGH poly A (L1-no bGH poly A), and used as a control for comparison with L2. The sequences were fused into the pBluescript II KS (+) plasmid with a T7 promoter (GENEWIZ, Suzhou, China). The details related to the sequences are given in Figure 1. In vitro transcription was performed for 6 h at 37 °C to produce the required RNA sequences. The transcribed RNA was then purified using the T7 RiboMAX Express Large Scale RNA Production System (P1320, Promega, Madison, WI, USA) and used for subsequent experiments.

### 2.3. Transfection

The 293T (CRL-11268™) and A549 (CRM-CCL-185) cells were purchased from American Type Culture Collection (VA, USA). Monolayer cells were cultivated in six-well plates in Dulbecco’s modified Eagle medium (DMEM) containing 10% fetal bovine serum and 1% penicillin and streptomycin in an incubator under 5% CO_2_ and 37 °C conditions. At 80% confluency, 5 μg L1 or L2 was added to each well for transfecting the cells using ER4000 transfection reagent, as per the manufacturer’s instructions (Engreen Biosystem, 4000-4, Beijing, China). The cells were then collected at relevant time points for subsequent experiments.

### 2.4. Luciferase Reporter Assay

The cell culture supernatant was collected at relevant time points for luciferase reporter assay; 20 μL supernatant and 50 μL fluorogenic substrate (QUANTI-Luc, rep-qlcg1, InvivoGen, San Diego, CA, USA) were mixed and luminescence was detected.

### 2.5. Real-time Quantitative PCR (qPCR)

Total RNA was extracted from cells using EasyPure RNA Kit (TransGen, P11108, Beijing, China). qPCR was performed to quantify gene expression using the One Step TB Green PrimeScript PLUS RT-PCR Kit (TaKaRa, AL40908A, Kyoto, Japan). *GAPDH* was used as a reference gene. Relative expression levels were calculated using the 2^−∆∆CT^ method. The primer sequences are listed in Table 1.

### 2.6. Western Blot (WB) Analysis

Total proteins were extracted by lysing the cells in RIPA buffer and quantitated using the BCA assay (Solarbio, PC0020, Beijing, China). Proteins were separated by electrophoresis on 8% SDS polyacrylamide gel (ExpressPlus™ PAGE Gel, 10×8, 8%, 10 wells, Genescript, Nanjing, China) and transferred onto nitrocellulose membranes. The membranes were blocked with 5% skimmed milk for 2 h at room temperature and then incubated with a primary antibody for 12 h at 4 °C. They were then washed three times with TBST and incubated with HRP-conjugated secondary antibody (ZSQB-BIO, ZB2305, Beijing, China) for 1 h at room temperature. The membranes were again washed three times with TBST. Finally, immunoblot results were visualized using ECL hypersensitive luminous solution (ThermoFisher, 32209, Waltham Mass, MA, USA) and detected using a CCD imaging system.

### 2.7. Vaccine Preparation

RBD protein used in this study was same as that of recombinant COVID-19 vaccine (ZF2001) donated by Anhui Zhifei Longcom Biopharmaceutical Co., Ltd., Hefei, China. Recombinant COVID-19 vaccine (ZF2001) was used for comparison (abbreviated RBD + Al). The doses of L2 used were 5, 10, 15, and 20 μg. The dose of RBD (one-fifth of the human dose) was 10 μg. The doses of L1, Al, CpG 7909, and poly(I:C) were 15, 100, 50, and 50 μg based on published studies [21,22,23]. L1 and L2 were encapsulated in lipid nanoparticles (LNP) donated by Beijing Minhai Biotechnology Co., Ltd., Beijing, China. Because CpG 7909 and poly(I:C) can enter cells through endocytosis, they do not need to be encapsulated in LNP. The adjuvants and RBD were thoroughly mixed and emulsified to prepare vaccines.

### 2.8. Animal Studies

*Animal study 1*: To investigate whether L2 could improve the immunogenicity of RBD compared with Al and to determine the optimal dose of L2, 36 Balb/c mice (6–8 weeks, 18–20 g) were randomly and equally assigned to six groups, namely Blank, RBD + Al, RBD + L2 (5 μg), RBD + L2 (10 μg), RBD + L2 (15 μg), and RBD + L2 (20 μg). The detailed experimental procedure is illustrated in Figure 2A.

*Animal study 2*: To determine whether L2 is more effective in improving the immunogenicity of RBD compared with CpG 7909 and poly(I:C), 54 Balb/c mice (6–8 weeks, 18–20 g) were randomly and averagely assigned to nine groups, namely Blank, RBD, L2, RBD + LNP, RBD + Al, RBD + L1, RBD + L2, RBD + CpG 7909, and RBD + poly(I:C). The dose of L2 (15 μg) optimized in *Animal study 1* was used in this study. The detailed experimental procedure is illustrated in Figure 2B.

*Animal study 3*: To compare whether L2 plus Al would be more effective in improving the immunogenicity of RBD compared with Al, L2, CpG plus Al, and poly(I:C) plus Al, 42 Balb/c mice (6–8 weeks, 18–20 g) were randomly and averagely assigned to seven groups, namely RBD + LNP + Al, RBD + L1 + Al, RBD + L2 (5 μg) + Al, RBD + L2 (10 μg) + Al, RBD + L2 (15 μg) + Al, RBD + CpG7909 + Al, and RBD + poly(I:C) + Al. The detailed experimental procedure is illustrated in Figure 2B.

### 2.9. Measurement of RBD-Specific IgG Titers Using ELISA

Levels of serum RBD-specific IgG titers were measured using ELISA. We coated 96-well EIA/RIA plates with 10 ng RBD protein per well at 4 °C for 12 h. ELISA was performed as described by us previously [24,25,26].

### 2.10. Serum Neutralization Assay

Neutralization assay for pseudo and authentic viruses was performed to measure neutralizing antibody (NAb) titers as described by us previously [24,27].

### 2.11. IFN-γ ELISPOT Assay

IFN-γ, secreted by splenocytes, was detected using the IFN-γ ELISPOT assay according to the method previously established by our research team [24,26,27]. The final levels were calculated by subtracting the background levels from the measured values.

### 2.12. Cytokine Assay

Freshly isolated splenocytes (3 × 10^5^ cells/ per well) were stimulated with the peptide pool spanning the RBD protein for 20 h. The concentration of each peptide was 5 μg/mL. The peptide pool was generated according to the method previously established by us [26]. After stimulation, the supernatant was collected from each well. Levels of IL-2, IL-6, and IL-10 were determined. All samples were assayed by Shanghai univ Biotechnology Co., LTD.

### 2.13. Statistical Analyses

Statistical analyses were performed using GraphPad Prism 8.0. Statistical significance was evaluated for at least three replicates using the Tukey’s multiple comparison test. A value of *p* < 0.05 indicates statistically significant difference.

## 3. Results

### 3.1. L1 and L2 Activated the RIG-I Signaling Pathways

IFN-β promoter relative fluorescence intensities induced by L1 and L2 were higher than those induced by L1-no bGH poly A and L2-no bGH poly A (Figure 3A). The relative *RIG-I* expression levels were substantially increased by L1 and L2, but those of *TLR-3*, *TLR-7*, *TLR-8*, myeloid differentiation primary response 88 (*MyD88*), and mitochondrial antiviral signaling protein (*MAVS*) were not increased in the A549 cells as determined using qPCR (Figure 3B). Expression levels of RIG-I, MyD88, and interferon-stimulated gene 56 (ISG56) were increased post-transfection with L1 and L2 compared with that in the ER4000 reagent control (Figure 3C). The relative fluorescence intensity of the 293T cells overexpressing *RIG-I* was dramatically increased, and was significantly higher than that of the 293T cells null, *RIG-I*^−/−^ 293T cells, *TLR-7*^−/−^ 293T cells, *TLR-8*^−/−^ 293T cells, and *MDA5*^−/−^ 293T cells at 24 and 48 h, post-transfection with L1 and L2, respectively. The relative fluorescence intensity was higher in the case of L2 than it was for L1 at 24 and 48 h (Figure 3D,E). These results indicate that bGH poly A could increase the activation level of L1 and L2. L1 and L2 could initiate RIG-I signal pathways to effectively activate the innate immunity, and the activation level of L2 was higher than that of L1.

### 3.2. L2 Improved the Immunogenicity of RBD Compared with Al

At 14 days after the first immunization and 14 days after the second immunization, RBD-specific IgG titers in the RBD + L2 (5 μg), RBD + L2 (10 μg), RBD + L2 (15 μg), and RBD + L2 (20 μg) groups were higher than in RBD + Al group (Figure 4A,C). NAb titers against pseudovirus in the RBD + L2 (5 μg), RBD + L2 (10 μg), RBD + L2 (15 μg), and RBD + L2 (20 μg) groups were also higher than in the RBD + Al group in a dose-dependent manner, and, in particular, the titers in the RBD + L2 (15 μg) group were significantly higher than in RBD + Al group (*p* < 0.05 and *p* < 0.001) (Figure 4B,D).

To further investigate whether L2 could more effectively improve the immunogenicity of RBD compared with Al, we determined the NAb titers against the authentic prototypes, Delta and Omicron. The NAb titers of RBD + L2 (5 μg), RBD + L2 (10 μg), RBD + L2 (15 μg), and RBD + L2 (20 μg) for the above three authentic SARS-CoV-2 strains were higher than that of RBD + Al in a dose-dependent manner, and, in particular, NAb titers of RBD + L2 (15 μg) were about 10-, 7-, and 4-fold, respectively, higher than that of RBD + Al. Moreover, the titer was the highest for the 15 μg dose compared with the titers obtained with other doses (Figure 4E–G). These results indicated that L2 could more effectively improve the immunogenicity of RBD in the dose range of 5 to 20 μg compared with Al. The optimal dose of L2 for inducing the highest NAb titers in mice was 15 μg.

### 3.3. L2 Improved the Immunogenicity of RBD Compared with CpG 7909 and Poly(I:C)

RBD-specific IgG titers in the RBD + L2 group were markedly higher than those in the Blank, RBD, LNP, RBD + Al, RBD + L1, RBD + CpG 7909, and RBD + Poly(I:C) groups at 14 days after the first immunization and at 14 days after the second immunization (*p* < 0.001 and *p* < 0.0001). There was no difference between L2 and RBD (*p* > 0.05; Figure 5A,B). NAb titers against the Delta and Omicron variants in the RBD + L2 group were also higher than those in the L2, RBD + Al, RBD + L1, RBD + CpG7909, and RBD + Poly(I:C) groups at 14 days after the second immunization, and, in particular, in the RBD + L2 group, the NAb titer against Omicron were increased by 8.68-fold compared with those in the RBD + Al group. The NAb titers against authentic viruses in the L2 group were below the minimum detection limit (Figure 5C,D). The above data indicate that L2 could improve the immunogenicity of RBD more effectively compared with CpG 7909 and poly(I:C). No difference between the L2 and RBD groups suggested that L2 cannot be translated into RBD peptide in mice.

### 3.4. L2 plus Al Improved the Immunogenicity of RBD Compared with CpG 7909 plus Al and Poly(I:C) plus Al

The RBD-specific IgG titers were higher in the RBD + L2 (15 μg) + Al group than those in the RBD + LNP + Al, RBD + CpG7909 + Al, RBD + Poly(I:C) + Al, and RBD + L1 + Al groups at 14 days after the first immunization and at 14 days after the second immunization (Figure 6A,B). Moreover, RBD-specific IgG titers and NAb titers against the pseudovirus in the L2 (5 μg) plus Al group were obviously higher than those in the L2 and Al groups (Figure A1A,B).

For the NAb titers against the authentic prototype, Delta and Omicron, those of RBD + L2 + Al were also higher than those of RBD + Al, RBD + CpG7909 + Al, RBD + Poly(I:C) + Al, and RBD + L2 to varying degrees, and, in particular, NAbs to Omicron increased about 44-fold compared with RBD + Al at 14 days after the second immunization (Figure 6C,D and Figure A1C,D). Above results indicated that L2 plus Al could also more effectively improve the immunogenicity of RBD compared with single L2 (RBD + L2), single Al, CpG7909 plus Al and Poly(I:C) plus Al as adjuvants.

### 3.5. L2 and L2 plus Al Elevated the Level of RBD-Specific Th1 Cell Immune Response Represented by IFN-γ Secretion

The levels of IFN-γ secreted by splenocytes in the RBD + L2 group were markedly higher than in the RBD + Al, RBD + CpG 7909, and RBD + Poly(I:C) groups (*p* < 0.01), but were lower than in the RBD + L1 group (*p* > 0.05; Figure 7A). The levels in RBD + L2 + Al group were higher than in the RBD + CpG7909 + Al and RBD + Poly(I:C) + Al groups, but lower than in the RBD + L1 + Al group (*p* > 0.05; Figure 7B). The levels of IL-2 (a Th1-type cytokine) in the RBD + L2 group were lower than in the RBD + L1 and RBD + Poly(I:C) group, and levels of IL-6 and IL-10 (Th2-type cytokines) in the RBD + L2 group were lower than in the RBD + Al, RBD + L1, RBD + CpG 7909, and RBD + Poly(I:C) group (Figure A2A–C). The levels of IL-2 in the RBD + L2 + Al group were lower than in the RBD + L1 + Al and RBD + Poly(I:C) + Al groups, and the levels of IL-6 and IL-10 in RBD + L2 + Al group were lower than in the RBD + L1 + Al, RBD + CpG7909 + Al, and RBD + Poly(I:C) + Al groups (Figure A2D–F). These results indicated that L2 and L2 plus Al could improve the levels of the RBD-specific Th1 immune response represented by IFN-γ secretion, rather than by the Th2 immune response in mice.

## 4. Discussion

The use of novel adjuvants is one of the effective approaches for improving the immunogenicity of recombinant protein vaccines [28,29]. In this study, we designed an ssRNA L2 originating from the SARS-CoV-2 prototype genome as an adjuvant. L2 could improve the immunogenicity of the SARS-CoV-2 recombinant RBD more effectively compared with the Al adjuvant and other nucleic acid-based novel adjuvants, such as CpG 7909 and poly(I:C).

The initiation of the PRR signal pathways plays an indispensable role in the activation of innate immunity [12,30]. We found that L2 could activate the RIG-I signal pathways post-transfection. The ssRNA containing the hairpin and the 5′ppp structures could activate the RIG-I signaling pathway in this study. We will further explore its detailed mechanism in the future. Moreover, we found that the bGH polyA sequence could increase the levels of nucleic acid activated-innate immunity. The published study indicated that bGH polyA could make the RNA structure stabilizable [31]. In this study, we speculated that bGH polyA might make the L2 adjuvant more stabilizable to activate the RIG-I signal pathway more effectively. We inserted the RBD sequence into the L2 sequence to adequately initiate the PRR signal pathways to activate the innate immunity more effectively, rather than translating into the RBD peptide. To improve the translation efficiency and lower the “self-adjuvant” effect without rapid degradation, the nucleic acid sequence of the mRNA vaccine was appended in the 5′-cap structure and embellished by using pseudouridine [32,33,34]. The L2 sequence, used as an adjuvant, was neither appended in the 5′-cap structure nor embellished using pseudouridine. Thus, L2 could not be translated into the RBD peptide, but could more effectively activate the innate immunity. Indeed, the absence of NAbs against authentic viruses in the L2 group confirmed that it cannot be translated into the RBD peptide.

Next, to check the RBD compatibility of L2, a SARS-CoV-2 recombinant protein vaccine was used and it was investigated whether L2 could improve the immunogenicity of RBD. ZF2001, a SARS-CoV-2 RBD recombinant protein vaccine approved with conditions in China, uses Al as the adjuvant (RBD + Al), and was used in this study for comparison [4]. RBD + LNP could not induce NAbs in mice, indicating that LNP used in this study does not act as an adjuvant. L2 was more effective in improving the immunogenicity of RBD in a dose-dependent manner over a range from 5 to 20 μg compared with ZF2001. At a dose of 15 μg, L2 was the most effective adjuvant compared with other doses in mice. The emergence of SARS-CoV-2 variants has resulted in the lowered efficacy of existing vaccines, affecting the global prevention and control effort for COVID-19 [35]. The Omicron variant is currently circulating as a VOC worldwide [3]. We obtained encouraging results that L2 (15 μg) used as an adjuvant for the recombinant protein vaccine prepared by us could increase NAb titers against authentic Omicron variants by about 8-fold compared with ZF2001. Dai et al. found that ZF2001 effectively preserved the neutralizing activity against newly emerging multiple variants [36]. In a previous study, we showed that ZF2001 could provide an effective protection against Delta variant-induced severe cases in a lethal model [37]. In this study, the result of L2 compatibility with RBD, inducing higher NAb titers, suggests that L2 can potentially be used as a novel adjuvant for the recombinant protein vaccine.

The molecular targeting for the PRR signaling pathways is one of the prospective directions in the research and development of novel adjuvants, such as poly(I:C) and CpG [8]. The published studies showed that CpG7909 and poly(I:C) could activate TLR-9 and TLR-3 signaling pathways, respectively [22,38,39]. The evaluation of the compatibility of the Coley/Pfizer HBV vaccine with CpG 7909 as an adjuvant is under a clinical trial [14]. In the present study, L2 could increase RBD-specific IgG titers of the vaccine to levels more than those achieved with CpG 7909 and poly(I:C). More intriguingly, L2 could also increase NAb titers against authentic Delta and Omicron variants by several folds compared with the titer achieved with poly(I:C) and even CpG 7909, which is already in a clinical trial. RBD-specific IgG titers and NAb titers achieved with L1, used as an adjuvant, were also lower than those achieved with L2. The reason for this may be that the longer sequence of L2 activated the innate immunity and subsequently induced B cell activation more efficiently. These results suggest that L2 could more effectively improve the immunogenicity of RBD as an adjuvant compared with poly(I:C) and CpG 7909.

The mechanisms underlying the role of Al as a vaccine adjuvant are that it activates the NOD-like receptor family pyrin domain containing 3 (NLRP3) signaling pathway and there is a slow and sustained release of the antigen to induce the production of antibodies continuously post immunization [39]. It has been shown that PRR agonists combined with Al as the adjuvant could provide higher immunogenicity compared with the Al adjuvant used alone. The evaluation of the compatibility of the NIAID malaria vaccine with CpG 7909 plus Al as adjuvants is being conducted in a clinical trial. CpG 7909 plus Al used as a composite adjuvant has also been used for the SARS-CoV-2 recombinant protein vaccine, which has been in clinical trial. In the present study, we found that the low dosage of L2 combined with Al used as the adjuvant could achieve higher immunogenicity of RBD compared with that achieved with L2 and poly(I:C) plus Al, and was even higher than that in the case of CpG 7909 plus Al. ZF2001 effectively preserved the neutralizing activity against the Alpha, Beta, Gamma, Delta, and Omicron variants [36,40,41]. In this study, it is encouraging that L2 plus Al used as a composite adjuvant could induce higher NAb titers than ZF2001 (RBD + Al); in particular, NAbs against Omicron increased about 44-fold.

Numerous studies have indicated that the Th1 immune response could induce the activation of the CD8^+^ T immune response, which plays a critical role in resisting SARS-CoV-2 invasion and in killing the virus by cytotoxic T lymphocytes (CTL) [42,43]. Inactivated viral vaccines and recombinant protein vaccines that cannot effectively induce the cellular immune response have always been a puzzle. Thus, the research and development of novel adjuvants can lead to strategies for effectively inducing the cellular immune response. The novel saponin adjuvant, Matrix-M™, was used for the SARS-CoV-2 recombinant protein vaccine, and it not only induced higher levels of the Th1 and Th2 immune response, but also activated the cytotoxic T-lymphocyte (CTL) response [44]. In this study, we found that ZF2001 could increase the secretion of RBD-specific IL-6 and IL-10 (Th2 type cytokines). However, IFN-γ and IL-2 are rarely secreted post immunization with ZF2001. On the contrary, L2 and L2 plus Al as adjuvants significantly improved the RBD-specific IFN-γ and IL-2 levels (Th1 type cytokines) compared with the levels obtained in the case of Al (ZF2001, with Al as an adjuvant), CpG 7909/plus Al, and poly(I:C)/plus Al in mice. Thus, we speculate that L2 and L2 plus Al used as adjuvants for the recombinant protein vaccine could improve the levels of Th1 to induce the CD8^+^ CTL mediated- killing of viruses.

The published studies indicate that the efficiency of activation of the PRR signaling pathways induced by nucleic acid sequences can be improved by optimizing the base sequences [12]. The optimization of the L2 base sequence is one of our future research projects to improve the efficacy of adjuvants more effectively. Subsequently, we will investigate whether it can offer protection against SARS-CoV-2 infection in animal models. The suitable delivery system is indispensable for the novel RNA adjuvants. RNA compatibility with others might obtain more efficacy, so RNA compound adjuvants are one of the research directions. As biological substances with the potential ability to be approved for the market, strict quality control is critical for novel RNA adjuvants with high-quality in industrial production. In addition, the safety of novel RNA adjuvants should be considered in the long term.

Overall, a novel ssRNA-based adjuvant L2 designed from the SARS-CoV-2 prototype genome could more effectively improve the immunogenicity of the RBD compared with Al and other nucleic acid-based adjuvants, namely CpG 7909 and poly(I:C). These findings should provide a basis for further research into and development of RNA-based novel adjuvants.

## Figures and Tables

**Figure 1 viruses-14-01854-f001:**
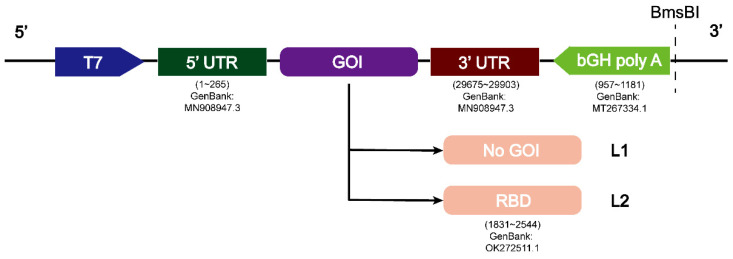
Structures of L1 and L2 containing the SARS-CoV-2 prototype genes. Numbers below the figure indicate the GenBank accession numbers of the gene sequences. GOI, gene of interest.

**Figure 2 viruses-14-01854-f002:**
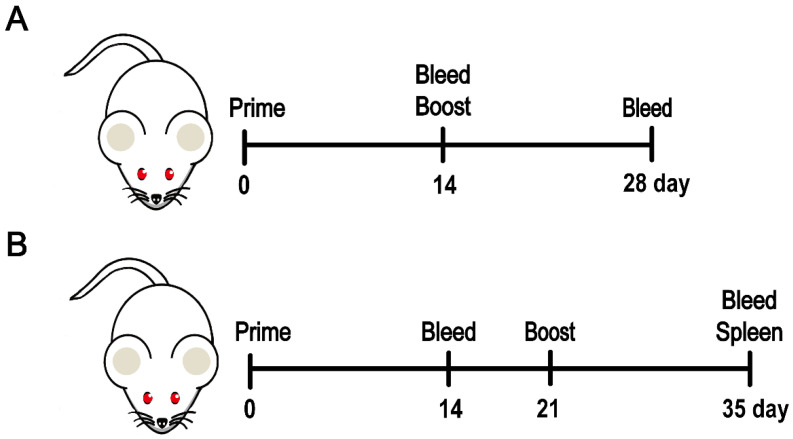
The detailed experimental procedure for the animal studies. (**A**) The animal study 1. (**B**) The animal studies 2 and 3.

**Figure 3 viruses-14-01854-f003:**
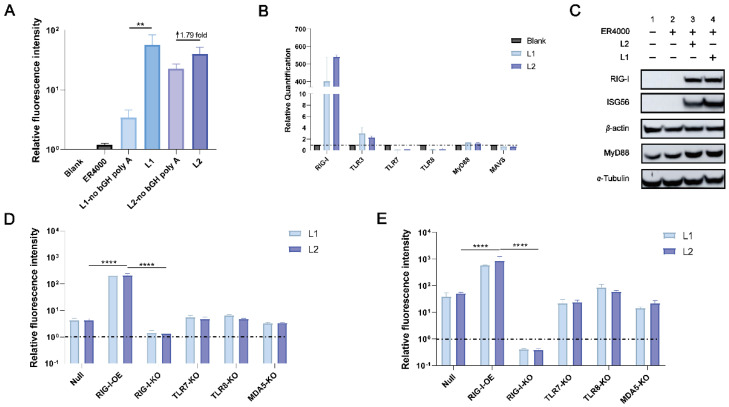
L1 and L2 activated the RIG-I signaling pathways in vitro. (**A**) The relative fluorescence intensity of Blank, ER4000, L1-no bGH poly A, L1, L2-no bGH poly A, and L2. (**B**) The relative expression of genes in Blank, L1, and L2 in A549. (**C**) Western blotting of A549 post-transfection with L1 and L2. Three independent experiments were repeated with similar results. (**D**,**E**) The relative fluorescence intensity of 293T null, 293T *RIG*-*I* overexpression (*RIG*-*I*-*OE*), *RIG*-*I*^−/−^ 293T, *TLR*-*7*^−/−^ 293T, *TLR*-*8*^−/−^ 293T, and *MDA5*^−/−^ 293T at 24 and 48 h post−transfection with L1 and L2. (**D**) 24 h. (**E**) 48 h. **, *p* < 0.01. ****, *p* < 0.0001. ↑, increase. ER4000 is a kind of transfection reagents purchased from Engreen Biosystem, Ltd., Beijing, China. in vitro experiments.

**Figure 4 viruses-14-01854-f004:**
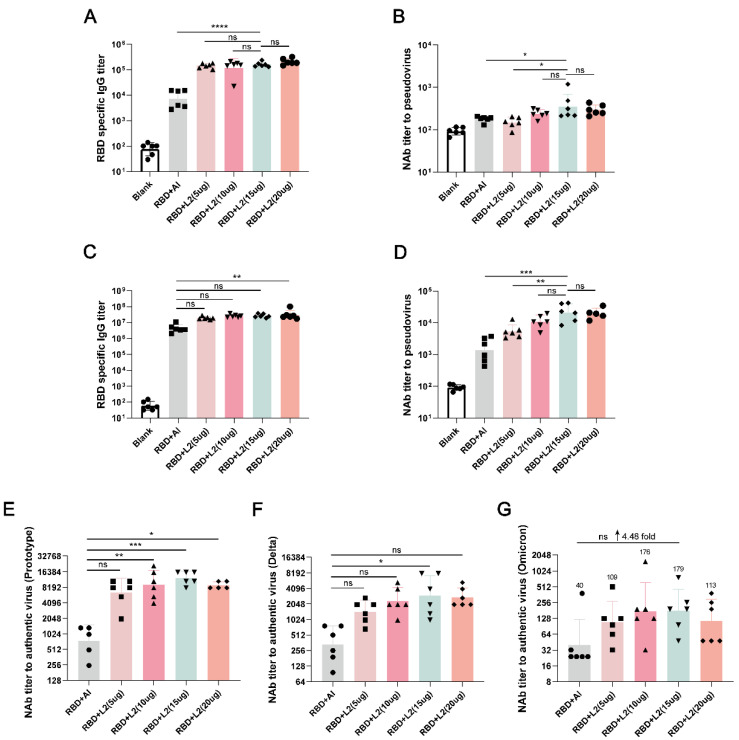
L2 improved the immunogenicity of the RBD compared with aluminum hydroxide (Al) adjuvant. (**A**,**B**) RBD-specific IgG titers and NAb titers against pseudovirus in Blank, RBD + Al, RBD + L2 (5 μg), RBD + L2 (10 μg), RBD + L2 (15 μg), and RBD + L2 (20 μg) groups at 14 days after the 1st immunization. (**A**) RBD-specific IgG titers. (**B**) NAb titers against pseudovirus. (**C**,**D**) RBD-specific IgG titers and NAb titers to pseudovirus in Blank, RBD + Al, RBD + L2 (5 μg), RBD + L2 (10 μg), RBD + L2 (15 μg), and RBD + L2 (20 μg) groups at 14 days after the 2nd immunization. (**C**) RBD-specific IgG titers. (**D**) NAb titers against pseudovirus. (**E**–**G**) NAb titers to authentic prototypes, Delta and Omicron, viruses in Blank, RBD + Al, RBD + L2 (5 μg), RBD + L2 (10 μg), RBD + L2 (15 μg), and RBD + L2 (20 μg) groups at 14 days after the 2nd immunization. (**E**) Prototype virus. (**F**) Delta variant. (**G**) Omicron variant. * *p* < 0.05. ** *p* < 0.01. *** *p* < 0.001. **** *p* < 0.0001. ↑, increase. *n* = 6 for each group.

**Figure 5 viruses-14-01854-f005:**
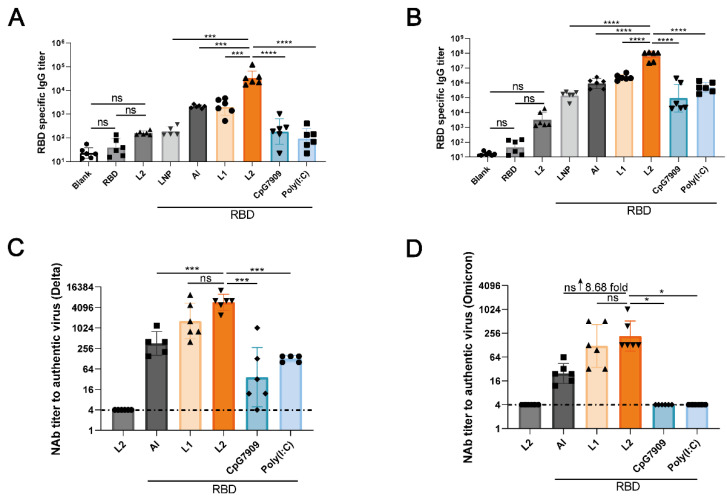
L2 improved the immunogenicity of the RBD compared with CpG 7909 and poly(I:C). (**A**,**B**) RBD-specific IgG titers in Blank, RBD, L2, RBD + LNP, RBD + Al, RBD + L1, RBD + L2, RBD + CpG7909, and RBD + Poly(I:C) groups at 14 days after the 1st immunization and at 14 days after the 2nd immunization. (**A**) At 14 days after the 1st immunization. (**B**) At 14 days after the 2nd immunization. (**C**,**D**) NAb titers against authentic Delta and Omicron variants in L2, RBD + Al, RBD + L1, RBD + L2, RBD + CpG7909, and RBD + Poly (I:C) groups at 14 days after the 2nd immunization. (**C**) Delta variant. (**D**) Omicron variant. * *p* < 0.05. *** *p* < 0.001. **** *p* < 0.0001. *n* = 6 for each group. RBD is the recombinant protein donated by Anhui Zhifei Longcom Biopharmaceutical Co., Ltd., Hefei, China.

**Figure 6 viruses-14-01854-f006:**
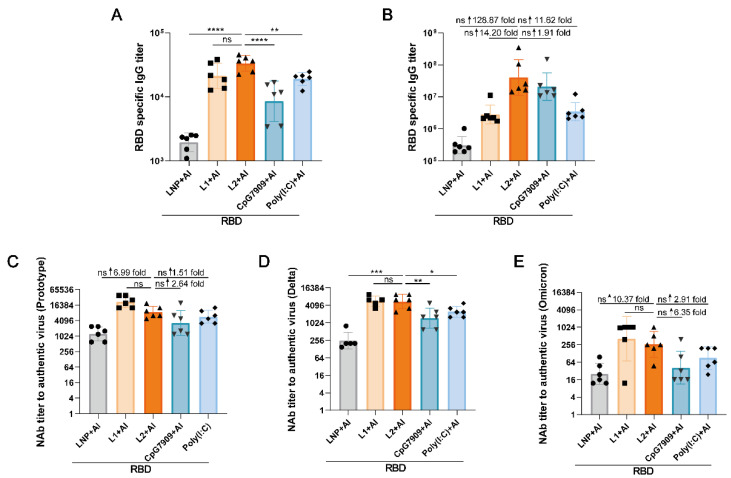
L2 + Al improved the immunogenicity of the RBD compared with CpG 7909 + Al and Poly(I:C) + Al. (**A**,**B**) RBD-specific IgG titers in RBD + LNP + Al, RBD + L1 + Al, RBD + L2 + Al, RBD + CpG7909 + Al, and RBD + Poly(I:C) + Al groups at 14 days after the 1st immunization and at 14 days after the 2nd immunization. (**A**) At 14 days after the 1st immunization. (**B**) At 14 days after the 2nd immunization. (**C**–**E**) NAb titers to authentic prototype, Delta and Omicron, variants in RBD + Al, RBD + L1 + Al, RBD + L2 + Al, RBD + CpG7909 + Al, and RBD + Poly(I:C) + Al groups at 14 days after the 2nd immunization. (**C**) Prototype. (**D**) Delta variant. (**E**) Omicron variant. * *p* < 0.05. ** *p* < 0.01. *** *p* < 0.001. **** *p* < 0.0001. ↑, increase. *n* = 6 for each group.

**Figure 7 viruses-14-01854-f007:**
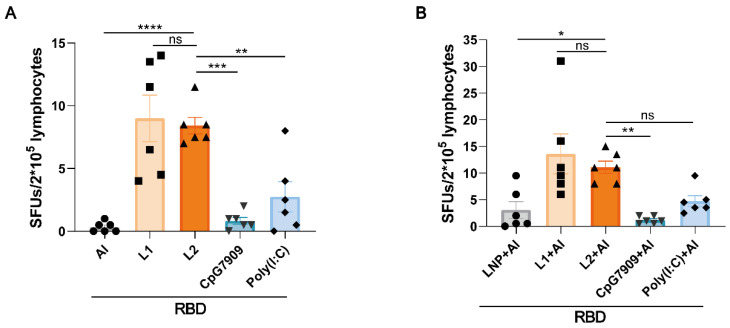
L2 and L2 + Al increased the levels of RBD-specific T-cell secreted IFN-γ immune response induced by the recombinant protein vaccine in mice. (**A**) The RBD-specific T cells secreted IFN-γ in RBD + Al, RBD + L1, RBD + L2, RBD + CpG 7909, and RBD + Poly(I:C) at 14 days after the 2nd immunization. (**B**) The spike-specific T cells secreted IFN-γ in RBD + LNP + Al, RBD + L1 + Al, RBD + L2 + Al, RBD + CpG7909 + Al, and RBD + Poly(I:C) + Al groups at 14 days after the 2nd immunization. * *p* < 0.05. ** *p* < 0.01. *** *p* < 0.001. **** *p* < 0.0001. *n* = 6 for each group.

**Table 1 viruses-14-01854-t001:** Primer sequences of qPCR.

Gene	Direction	Sequence (5′-3′)
*RIG-I*	Forward Reverse	GATTATATCCGGAAGACCCTG ATACTGCACCTCTTCCTCC
*TLR-3*	Forward Reverse	TCATCCAACAGAATCATGAGAC CTAACAGTGCACTTGGTGG
*TLR-7*	Forward Reverse	CAAGAAAGTTGATGCTATTGGG GTGTCCACATTGGAAACAC
*TLR-8*	Forward Reverse	AGAAACATGGTTCTCTTGACAC TAGTCTCCTTTCCCAGGCT
*MyD88*	Forward Reverse	CAGCATTGAGGAGGATTGC GGGACACTGCTGTCTACAG
*MAVS*	Forward Reverse	CAGCAGAAATGAGGAGACC CTATTCTCAGAGCTGCTGTC
*GAPDH*	Forward Reverse	GGAGCGAGATCCCTCCAAAAT GGCTGTTGTCATACTTCTCATGG

## Data Availability

All data generated or analyzed and materials during this study are included in this published article.

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
