# Peer review of "A Novel Single-Stranded RNA-Based Adjuvant Improves the Immunogenicity of the SARS-CoV-2 Recombinant Protein Vaccine"

_viruses, 2022, doi:10.3390/v14091854_

Round 1

Reviewer 1 Report

1.Figure 3A, Figure 4A and Figure 5A appeared before Figure 2. Please modify it. The pictures should be set in order.

2.We could observe that bGH could increase the relative fluorescence intensity. Thus, please try to explain the reason in the discussion part.

3.In Figure 2 to Figure 6, and Figure S1 and S2, although some comparisons were not different, please mark “ns”, so as not to make readers mistakenly think that there were no comparisons.

4.Figure 4A and 5A, font styles were not uniform, please revise them to make figures aesthetics.

5.This is an application study that might be approved as a novel adjuvant in the future. But it also has some problems need to solve, please try to bring up a few problems that need to be solved in the discussion part. This will contribute to the research and development of novel RNA-based adjuvants. For example, the quality control.

Author Response

Responses to reviewers’ comments for the manuscript (NO.1812017)

Dear Editor/ reviewer,

We are very grateful to editors and reviewers for the interest in our work and those valuable advices and comments. We have adopted the suggestion and revised them accordingly. We invited the professional language service company help us to modify our manuscript. The revised parts were marked in green in the revised manuscript.

Reviewer #1 Comments

1.Figure 3A, Figure 4A and Figure 5A appeared before Figure 2. Please modify it. The pictures should be set in order.

Response: Thanks for this comment. In the revised manuscript, we descripted the animal study processes in a figure (Figure 2) and adjusted figure numbers.

2.We could observe that bGH could increase the relative fluorescence intensity. Thus, please try to explain the reason in the discussion part.

Response: Thanks for this helpful comment. In the revised manuscript, we added the discussion to try to explain the reason. “The published study indicated that bGH polyA could make the RNA structure stabilizable. In this study, we speculated that bGH polyA might make the L2 adjuvant more stabilizable to activate RIG-I signal pathway more effectively” (Line 333-336, Page 10)

Reference: Sakaguchi, M., Sadahira, T., Ueki, H., et al. Robust cancer-specific gene expression by a novel cassette with hTERT and CMV promoter elements[J]. Oncol Rep. 2017, 38(2): 1108-1114.

3.In Figure 2 to Figure 6, and Figure S1 and S2, although some comparisons were not different, please mark “ns”, so as not to make readers mistakenly think that there were no comparisons.

Response: Thanks for this advice. We marked “ns” in the revised manuscript.

4.Figure 4A and 5A, font styles were not uniform, please revise them to make figures aesthetics.

Response: We feel apologized for this neglect. We modified it in the revised manuscript.

5.This is an application study that might be approved as a novel adjuvant in the future. But it also has some problems need to solve, please try to bring up a few problems that need to be solved in the discussion part. This will contribute to the research and development of novel RNA-based adjuvants. For example, the quality control.

Response: Thanks for this helpful advice. In the revised manuscript, we added the discussion to try to bring up a few problems that need to be solved. “The suitable delivery system is indispensable for the novel RNA adjuvants. RNA compatibility with others might get the more efficacy, so RNA compound adjuvants are one of the research directions. As a biological substance with the potential ability to approve market, the strict quality control is critical for novel RNA adjuvants with high-quality in industrial production. In addition, the safety of novel RNA adjuvants should be concerned in the long term.” (Line 415-421, Page 12)

We really appreciate for the earnest work of Editors and Reviewers, and hope that the corrections will meet with approval. Once again, thank you very much for your comments and suggestions. We look forward to your information about my revised manuscript and thank you for your good comments.

Best regards!

Yours sincerely,

Dr. Zhenglun Liang

National Institutes for Food and Drug Control. Huatuo Road No.31, Daxing District, Beijing, China. 102629

Email: lzhenglun@126.com

Reviewer 2 Report

The manuscript is interesting, but the results shown are basically phenomenological and the possible reason for the results are not fully explained. The followings are questions for each Figures.

Figure 1. Structures of L1 and L2. I understand the transcripts were used as adjuvants for SARS-CoV-2 recombinant protein vaccine. In addition, what does bGH stand for? Bovine growth hormone? “bGH poly A signal” are often used for plasmid vectors. Are “bGH” and “poly A” different components in Figure 1?  Which properties of RBD and bGH are attributable to RIG-1 signaling pathway activation? 

Figure 2. Do you also examine CpG7909 or poly(I:C) effects on activation of RIG-I and TLR signaling pathway? If so, show them. Poly (I:C) is supposed to activate TLR3 signal pathway. What is ER4000 control?

Figure 3. The explanation of RBD used for the experiment is not enough. Authors only described that RBD was used for preparing the SARS-CoV-2 recombinant protein vaccine (page 4, lines 141,142). What is the structure? 

Figure 4. Why RBD specific IgG titer of L2 alone was higher than that of RBD? Doesn’t L2 alone contain RBD, does it? How many amounts of RBD, L1, L2, CpG7909, poly(I:C) were used for the experiments? In Figure 4E, authors stated that Nab titer against Omicron were increased by 8.68-fold compared with that in RBD+Al group (page 7, lines 253, 254). Is that statistically significant? Asterisk is not shown in Figure4E.

Figure 5. In Figure 5E, no asterisks are shown. Aren’t they statistically significant?

Other comments.

Line 146. Liposome and LNP are not the same. They are different.

Line 153. “Balb/C” →”Balb/c”.

The style of References is not correct. For example, #34, #35, #36 are not correctly written (formats of the authors’ name and journal name are not correct). Check them.

Author Response

Responses to reviewers’ comments for the manuscript (NO.1812017)

Dear Editor/ reviewer,

We are very grateful to editors and reviewers for the interest in our work and those valuable advices and comments. We have adopted the suggestion and revised them accordingly. We invited the professional language service company help us to modify our manuscript. The revised parts were marked in green in the revised manuscript.

Reviewer #2 Comments

  • Figure 1. Structures of L1 and L2. I understand the transcripts were used as adjuvants for SARS-CoV-2 recombinant protein vaccine. In addition, what does bGH stand for? Bovine growth hormone? “bGH poly A signal” are often used for plasmid vectors. Are “bGH” and “poly A” different components in Figure 1? Which properties of RBD and bGH are attributable to RIG-1 signaling pathway activation?

Response: Thanks for this helpful comment. bGH is the bGH poly A signal. We modified it in the whole manuscript. “The ssRNA containing the hairpin and the 5’ppp structures could activate RIG-I signaling pathway in this study. We will further explore its detailed mechanism in the future.” (Line 330-332, Page 10).

  • Figure 2. Do you also examine CpG7909 or poly(I:C) effects on activation of RIG-I and TLR signaling pathway? If so, show them. Poly (I:C) is supposed to activate TLR3 signal pathway. What is ER4000 control?

Response: Thanks for this comment. CpG7909 and poly(I:C) are two kinds of novel nucleic acid adjuvants that are approved market. The published studies showed that CpG7909 and poly(I:C) could activate TLR-9 and TLR-3 signaling pathways, respectively. (Line 367-368, Page 11). Thus, we used CpG7909 and poly(I:C) as the comparisons that are benefits for investigating the efficacy of the adjuvant that we designed in this study. ER4000 is a kind of transfection reagents in vitro experiments. We added the descriptions in the revised manuscript. (Line 221-222, Page 6)

Reference: Hu S., Chen H., Ma J., Chen Q., Deng H., Gong F., et al. CpG7909 adjuvant enhanced immunogenicity efficacy in mice immunized with ESAT6-Ag85A fusion protein, but does not confer significant protection against Mycobacterium tuberculosis infection. J Appl Microbiol. 2013 Nov;115(5):1203-1211.

Sabbaghi A., Malek M., Abdolahi S., Miri S. M., Alizadeh L., Samadi M., et al. A formulated poly (I:C)/CCL21 as an effective mucosal adjuvant for gamma-irradiated influenza vaccine. Virol J. 2021 Oct 9;18(1):201.

Nanishi E., Borriello F., O'Meara T. R., McGrath M. E., Saito Y., Haupt R. E., et al. An aluminum hydroxide: CpG adjuvant enhances protection elicited by a SARS-CoV-2 receptor-binding domain vaccine in aged mice. Sci Transl Med. 2021 Nov 16: eabj5305.

  • Figure 3. The explanation of RBD used for the experiment is not enough. Authors only described that RBD was used for preparing the SARS-CoV-2 recombinant protein vaccine (page 4, lines 141,142). What is the structure?

Response: Thanks for this helpful comment. In the revised manuscript, we added the explanation of RBD. “RBD used in this study was same as that of ZF2001 SARS-CoV-2 recombinant RBD protein vaccine. The GenBank number of the RBD sequence is YP_009724390 in NCBI.” (Line 100-102, Page 3)

Reference:       Dai L., Zheng T., Xu K., Han Y., Xu L., Huang E., et al. A Universal Design of Betacoronavirus Vaccines against COVID-19, MERS, and SARS. Cell. 2020 Aug 6;182(3):722-733 e711.

  • Figure 4. Why RBD specific IgG titer of L2 alone was higher than that of RBD? Doesn’t L2 alone contain RBD, does it? How many amounts of RBD, L1, L2, CpG7909, poly(I:C) were used for the experiments? In Figure 4E, authors stated that Nab titer against Omicron were increased by 8.68-fold compared with that in RBD+Al group (page 7, lines 253, 254). Is that statistically significant? Asterisk is not shown in Figure4E.

Response: Thanks for this helpful comment. RBD specific IgG titer of L2 alone was higher than that of RBD, but there is no difference between L2 alone and RBD. L2 alone contain RBD sequence, but it could not translate into RBD peptide. The amounts of RBD, L1, L2, CpG7909 and poly(I:C) was described in the material and method part in the revised manuscript. (Line 148-149, Page 4). That is no statistically significant between RBD + Al and RBD + L2 groups and we marked “ns” in Figure4E.

  • Figure 5. In Figure 5E, no asterisks are shown. Aren’t they statistically significant?

Response: Thanks for this helpful comment. In Figure 5D of the revised manuscript (Figure 5E of the manuscript), we compared L2 + Al with other groups to observe the efficacy of L2 + Al as the composite adjuvant. We marked the difference analysis in the revised manuscript.

  • Line 146. Liposome and LNP are not the same. They are different.

Response: Thanks for this helpful comment. Indeed, liposome and LNP are not the same. LNP is the abbreviation of the lipid nanoparticles We modified it in the revised manuscript. (Line 150, Page 4).

  • Line 153. “Balb/C” →”Balb/c”.

Response: Thanks for this comment. We modified it in the whole revised manuscript. (Line 156, 161 and 168, Page 4).

  • The style of References is not correct. For example, #34, #35, #36 are not correctly written (formats of the authors’ name and journal name are not correct). Check them.

Response: We feel apologized for this neglect. We modified it in the revised manuscript. (Line 540-548, Page 15)

We really appreciate for the earnest work of Editors and Reviewers, and hope that the corrections will meet with approval. Once again, thank you very much for your comments and suggestions. We look forward to your information about my revised manuscript and thank you for your good comments.

Best regards!

Yours sincerely,

Dr. Zhenglun Liang

National Institutes for Food and Drug Control. Huatuo Road No.31, Daxing District, Beijing, China. 102629

Email: lzhenglun@126.com

Round 2

Reviewer 2 Report

The manuscript has been improved after revision.

However, the manuscript has room to be revised. Followings are comments.

Line 87. “aluminum hydroxide (Al)” reads “Al” (See line 64).

Line 100. “RBD used in this study was same as that of ZF2001 SARS-CoV-2 recombinant RBD protein vaccine.” The sentence is not clear.

What does “that” of ZF2001 mean?

Lines 154, 155. “L1 and L2 were encapsulated in liposome to lipid nanoparticles (LNP)”.

This description seems to be incorrect. LNP and liposome are different. LNP does not include liposome!

Line 238. “ER4000 is a kind of transfection reagents in vitro experiments.”

The sentence is not clear. What is the company name selling “ER4000”?

Line 284. Why “L2” alone is not described here?

Line 318. “increase” reads “increased”.

Line 375. “L2/plus Al” reads “L2 plus Al”. 

Figure 3D, 3E. What does “RIG-I-OE” stand for?

Figure 5A, 5B. Does “RBD” described between “Blank” and “L2” mean RNA encoding RBD? Is this statement correct? Is it a protein or RNA? If it is a protein, it should be located above the RBD horizontal line.

“RBD” under the horizontal line means “RBD” protein, doesn’t it? It should be described clearly. 
